# Association between Work Environments and Stigma towards People with Schizophrenia among Mental Health Professionals in Japan

**DOI:** 10.3390/healthcare9020107

**Published:** 2021-01-21

**Authors:** Yuichi Kato, Rie Chiba, Sosei Yamaguchi, Kyohei Goto, Maki Umeda, Yuki Miyamoto

**Affiliations:** 1Department of Nursing, Graduate School of Health Sciences, Kobe University, Kobe, Hyogo 654-0142, Japan; crie-tky@umin.ac.jp; 2Department of Community Mental Health & Law, National Institute of Mental Health, National Center of Neurology and Psychiatry, Kodaira, Tokyo 187-8553, Japan; sosei.yama@ncnp.go.jp; 3Department of Psychiatric Nursing, Kyoritsu Women’s University, Chiyoda-ku, Tokyo 101-0051, Japan; whimsical-angels@umin.ac.jp; 4Research Institute of Nursing Care for People and Community, University of Hyogo, Akashi 673-8588, Japan; makiumeda-tky@umin.ac.jp; 5Department of Psychiatric Nursing, Graduate School of Medicine, The University of Tokyo, Bunkyo-ku, Tokyo 113-0033, Japan; yyuki-tky@umin.ac.jp

**Keywords:** control over practice, mental health professionals, stigma, work environment

## Abstract

This study aimed to examine the association between control over practice in work environments and stigma toward people with schizophrenia among mental health professionals. We conducted secondary analyses on data from a self-administered questionnaire survey. The sample in the initial study included mental health professionals from two psychiatric hospitals, 56 psychiatric clinics, and community service agencies in Japan. The Ethics Committee of the University of Tokyo, approved this study. Data from 279 participants were used for secondary analyses (valid response rate = 58.7%). The hierarchical multiple regression analysis was used to determine the association between control over practice and stigma. We performed subgroup analyses among nurses (*n* = 121) and psychiatric social workers (*n* = 92). Control over practice was negatively associated with stigma among mental health professionals (β = −0.162, *p* < 0.01). The subgroup analyses among nurses indicated that control over practice, educational history and recovery knowledge were associated with stigma. However, these variables were not associated with stigma among psychiatric social workers. Control over practice might help to reduce stigma among mental health professionals. Factors related to stigma might differ by occupation. Therefore, further comprehensive studies among various professionals would further our understanding of these factors.

## 1. Introduction

Overcoming mental health-related stigma is a global issue as it adversely influences all aspects of the lives of people with mental illnesses [1,2]. The concept of mental health-related stigma in the general public is categorized into three components: problems of knowledge (ignorance), attitudes (prejudice), and general behavior (discrimination) [1]. Among various mental illnesses, schizophrenia is one of the major and common illnesses. With one contributing factor that considering schizophrenia as a genetic disorder [3,4], schizophrenia often is considered as a genetic disorder, it is more likely to be associated with negative beliefs, social distance, and discrimination than other mental illnesses such as depression or bipolar disorders [5,6]. Earlier systematic reviews state that professionals suffer mental health-related stigma and identified them as among those that endure severe stigma [7,8,9,10]. Therefore, mental health professionals are regarded as a target group for anti-stigma intervention.

Mental health-related stigma may affect the quality of care and services meant to support personal recovery [11,12,13]. Personal recovery is a widely accepted central concept in mental health care [14,15]. Personal recovery is described as “a deeply personal, unique process of changing one’s attitude, values, feelings, goals, skills, and/or roles. It is a way of living a satisfying, hopeful, and contributing life even within the limitations caused by illnesses” [16]. In Japan, previous studies stated that staffs in psychiatric wards as well as community care staffs had stigma towards people with mental illness [17,18]. Additionally, past studies show that mental health-related stigma among professionals could have some negative impacts on their’ mental health, such as increasing the risk of burnout and reducing job satisfaction among others [19,20]. Therefore, it is important to develop strategies to reduce stigma among mental health professionals as mental health-related stigma affects their mental health as well as the quality of care they offer. However, most of the previous studies aimed at overcoming mental health-related stigma in the general public rather than among professionals and, therefore, paid less attention to the stigma among professionals. [10,21,22,23,24].

Previous studies theoretically suggest that the key factors for reducing mental health-related stigma among professionals would be ensuring safety in their workplaces, and creating organizational cultures which encourage staffs to adopt the care to support a patient’s personal recovery, as well as the accumulation of positive experiences through such care and evidence-based practices [25,26,27] and the improvement of recovery knowledge which refers to the accurate knowledge and attitudes towards recovery-oriented practices [28]. Therefore, work environments surrounding mental health professionals might be crucial to reducing stigma towards people that suffer mental illnesses. However, to date, no studies have investigated the association between stigma and workplace practice environments that support daily practices among mental health professionals. Workplace practice environments as one of the work environments that support the daily practices of mental health professionals could create organizational cultures and ensure safety which could help to reduce stigma. Moreover, several past studies state that attitudes toward people with mental illness differ among psychologists, psychiatrists, and psychiatric nurses [29,30,31,32]. According to the study conducted by Serafini et al. (2011), medical students and doctors expressed higher levels of stigma towards people with schizophrenia than psychiatric nurses [3]. Therefore, the factors that lead to stigma may vary by profession. However, the relevant evidence is still insufficient.

Workplace practice environments have some aspects such as control over practice, leadership, staff relationships with physicians, and teamwork [33]. Among them, control over practice in workplace practice environments signifies sufficient intra-organizational status to influence others and deploy resources, including abundant human resources, appropriate support, and sufficient time to be involved with patients [34]. High control over practice enables mental health professionals to practice good care with intra-organizational resources. This positive experience might be related to their mental health as well as positive attitude and lower stigma towards people with schizophrenia. However, no studies have examined the association between control over practice as one of the workplace environments and stigma among mental health professionals. Therefore, this study aimed to examine the association between control over practice in workplace practice environments and stigma towards people with schizophrenia among mental health professionals in Japan. In particular, we hypothesized that control over practice in workplace practice environments would be negatively related to stigma towards people with schizophrenia, among mental health professionals in Japan. Clarifying the relationship between stigma towards people with schizophrenia and workplace practice environments surrounding professionals would offer novel insights into the factors that would help to reduce stigma among mental health professionals. Additionally, this study used subgroup analyses to examine whether the respective factors differed by occupation, for better understanding of the factors.

## 2. Materials and Methods

### 2.1. Participants

We conducted a secondary analysis on data using a self-administered questionnaire survey from February to March 2012 [28,35]. Both full-time and part-time professionals with the following occupations were eligible for inclusion: psychiatrists, registered/assistant nurses, public health nurses, clinical psychologists, pharmacists, occupational therapists, and psychiatric social workers. Participants were recruited from two psychiatric hospitals, and 56 psychiatric clinics and community service agencies in the Kanto region in Japan. The two psychiatric hospitals had 220 eligible mental health professionals. Among them, 180 agreed to participate and returned completed questionnaires. In the psychiatric clinics and community service agencies such as day-care facilities and employment support service facilities, 255 eligible professionals were identified. Among them, 151 agreed to participate and responded to the questionnaire. Therefore, a total of 331 responses were obtained; however, 52 were excluded because of missing responses for one or more of the items on either the Japanese-language version of the Social Distance Scale (SDSJ) or control over practice as a sub-category of the Revised Professional Practice Environment (RPPE) Scale. In conclusion, we used data from the remaining 279 participants to conduct the secondary analyses (valid response rate = 58.7%).

### 2.2. Measures

#### 2.2.1. Japanese-Language Version of Social Distance Scale (SDSJ)

The Social Distance Scale (SDS) was developed based on the original scale created by Whatley, which is a five-item scale designed to assess an individual’s sense of social distance from people with schizophrenia [36]. SDS includes items such as “I think it is best not to associate with people with schizophrenia”, “It would bother me to live near a person with schizophrenia who had been in a mental hospital”, and “I would not ride in a taxi driven by someone with schizophrenia who had been in a mental hospital”. Responses are rated on a four-point Likert scale ranging from 0 (disagree) to 3 (agree), with a higher score exhibiting a more negative attitude. SDSJ has been found to have good internal consistency (Cronbach’s α = 0.87) and good test-retest reliability and acceptable factorial validity [37].

#### 2.2.2. Control over Practice

The Revised Professional Practice Environment (RPPE) Scale was developed to evaluate the effectiveness of workplace practice environments in supporting professionals in their delivery of patient care [33]. In this study, we used the items of control over practice domain among eight domains of the RPPE scale [33]. It consists of five items, such as “I have adequate support services to allow me to spend time with my patients,” “We have enough staff to get the work done,” and “There are enough staff to provide quality patient care.” Responses are rated on a four-point Likert scale ranging from 1 (strongly disagree) to 4 (strongly agree). Higher scores indicate greater control over practice. The RPPE scale was found to have good construct validity while the subscale exhibited good reliability (Cronbach’s α = 0.84) [34]. We used the total score of the five-item subscale in this study.

#### 2.2.3. Other Variables and Demographic Data

A trichotomous question that asked whether participants knew about the word of personal recovery was included in the questionnaire. Occupational and demographic data, including settings (psychiatric wards or facilities in the community), employment status (full-time or part-time job), age, sex, and educational history were also obtained in the initial study.

#### 2.2.4. Statistical Analysis

We used descriptive analyses to describe the occupational and demographic characteristics of this sample. We further used the Mann-Whitney U test to compare the mean scores of the SDS between subgroups, namely nurse and psychiatric social workers, as SDSJ scores were abnormally distributed. We confirmed that multiple regression analyses were applicable based on the results of the QQ plot and Durbin–Watson ratio in a single regression analysis between the SDSJ and control over practice. To test our hypothesis that control over practice in work environments would be negatively related to the SDSJ score, we further used hierarchical multiple regression analyses with the forced entry method. First, we conducted a single regression analysis to examine the relationship between the SDSJ as the dependent variable and control over practice as the independent variable (step 1). Demographic variables (i.e., age and education level; step 2), settings (step 3), and recovery knowledge (step 4) were entered as control variables. We entered the settings prior to recovery knowledge in order to examine the relevance of recovery knowledge to stigma, after excluding the effects of settings, including differences in patient characteristics, education, and treatment philosophy caused by various work settings. Settings (i.e., psychiatric wards vs. facilities in community, outpatient clinics, or psychiatric day-cares) and education level (i.e., four-year college degree or above, or less) were used as dummy variables. We also conducted subgroup analyses to examine whether the factors related to stigma differed between nurses and psychiatric social workers, which had more than 30% of the total participants. We further used the variance inflation factor (VIF) to check the data for multicollinearity. We used the Durbin–Watson ratio to check the randomness of the residuals after conducting hierarchical multiple regression analyses. *p* values of less than 0.05 were considered statistically significant. All the tests were two-tailed. All statistical analyses were performed using the SPSS statistical software version 25 for Windows (IBM, Armonk, NY, USA).

#### 2.2.5. Ethical Consideration

This study was implemented as a secondary analysis of the initial study. The ethical consideration was described in another study [28,35]. The aims and procedures of the study were approved by the Ethical Committee of The University of Tokyo, Japan (No. 3607).

## 3. Results

### 3.1. Participants’ Characteristics and Outcome Scale Score

Table 1 shows the characteristics of the participants. The occupations included in this study were nurses (*n* = 121; 43.4%), social workers (*n* = 92; 33.0%), clinical psychologists (*n* = 20; 7.2%), occupational therapists (*n* = 18; 6.5%), psychiatrists (*n* = 13; 4.7%), social welfare counselors (*n* = 7; 2.5%), pharmacologists (*n* = 5; 1.8%), and public health nurses (*n* = 3, 1.1%). Of these, 43.4% worked in inpatient psychiatry. Table 1 presents the occupational and demographic characteristics of the participants, subgroups for nurses, and psychiatric social workers. The mean age of the participants was 40.7 years (range: 22–75 years; SD = 11.9); 70.6% were female. The mean length of tenure at the facilities was 9.9 years (range: 0–45 years; SD = 8.5). Most of the nurses worked in inpatient psychiatric wards (*n* = 101; 83.5%). On the other hand, most psychiatric social workers worked in community facilities (*n* = 89; 96.7%), especially employment support service facilities (*n* = 52; 56.5%). Psychiatric social workers (78.3%) were more likely to know the word of personal recovery than nurses (34.7%). 

SDSJ mean scores of the total participants were 5.86 ± 3.42. The SDSJ mean scores among the nurses and psychiatric social workers were 7.50 ± 3.32 and 3.88 ± 2.84, respectively. We found a significant difference between these occupation (MD = 3.62, 95%CI [2.33–4.91], *p* < 0.001). Additionally, the SDSJ score of mental health professionals in community facilities (M = 4.56, SD = 2.96) was lower than those in psychiatric wards (M = 7.56, SD = 3.23).

### 3.2. Association between Control over Practice and Stigma among all the Total Participants

Table 2 shows the results of the hierarchical multiple regression analyses among the total participants. The acceptable Durbin–Watson ratio and QQ plots confirmed the normality and randomness of the residuals. (i.e., the Durbin–Watson ratio was 1.470, and the QQ plot showed a nearly straight line.) The single regression analysis showed a significant and negative association between the SDSJ score and control over practice (β = −0.22, *p* < 0.01). When we added the demographics (i.e., age and educational history) to Model 1 as Model 2, a significant and negative association between the SDSJ score and control over practice were also exhibited (β = −0.20, *p* < 0.01) with statistical significance of the model (Adjusted R^2^ = 0.137, F = 15.69, *p* < 0.001). Then, we added the settings to Model 2 as Model 3, and a significant and negative association between the SDSJ score and control over practice still showed (β = −0.18, *p* < 0.01), with statistical significance of the model (Adjusted R^2^ = 0.219, F = 20.47, *p* < 0.001). Finally, we added the recovery knowledge to Model 3 as Model 4, and control over practice was still negatively associated with stigma towards people with schizophrenia (β = −0.16, *p* < 0.01). The final model (Model 4) was confirmed to be statistically significant (Adjusted R^2^ = 0.267, F = 21.21, *p* < 0.001). Additionally, all VIF values did not exceed 2.0 in this model. The Durbin–Watson ratio was 1.903 in this model.

### 3.3. Association between Control over Practice and Stigma in Nurses and Psychiatric Social Workers

We also conducted subgroup analyses to examine whether the factors related to stigma differed between nurses and psychiatric social workers. Table 3 and Table 4 show the results of the subgroup analyses. For nurses, control over practice (β = −0.26, *p* < 0.01) and recovery knowledge (β = −0.32, *p* < 0.001) were negatively associated with stigma, while educational history (β = 0.22, *p* < 0.05) was positively related to stigma (Table 3). The model was confirmed to be statistically significant (Adjusted R^2^ = 0.210, F = 7.37, *p* < 0.001). On the other hand, only settings (β = −0.29, *p* < 0.01) were related to stigma with regard to psychiatric social workers (Table 4). In other words, working in psychiatric wards was related to higher stigma. However, the model was not significant (adjusted R^2^ = 0.045, F = 1.86, *p* = 0.46). 

## 4. Discussion

The results of the hierarchical multiple regression analysis for all the participants of mental health professionals showed that control over practice was negatively associated with the SDS score. The same result was observed after adjusting for potential covariates. The subgroup analyses for nurses revealed that control over practice and recovery knowledge were negatively associated with the SDS score, whereas educational history was positively related to the SDS score. However, these variables were insignificantly related to the SDS score among psychiatric social workers.

As we hypothesized, the findings of the main analysis showed that control over practice was negatively associated with stigma among mental health professionals. Higher control over practice is obtained through practicing good care with adequate intra-organizational resources such as human resources [33,34]. Intra-organizational resources, which are one of the key elements in control over practice [33] are capable of improving mental health [38]. On the other hand, Zaninotto et al. (2018) suggested that mental health problems among mental health professionals such as low personal accomplishment at work would contribute to stigma [27]. One can argue that high control over practice might help improve mental health in professionals by practicing good care with intra-organizational resources, and thereby contribute to reducing stigma towards people with schizophrenia. Further studies should clarify this mechanism in the future. As previous studies have suggested [39,40], recovery knowledge was negatively associated with stigma among mental health professionals in this study. This finding suggests that education, which improves recovery knowledge might be important to reducing stigma among mental health professionals. Although it has been becoming clear that recovery knowledge is an important educational perspective for reducing stigma among professionals [41,42], there is no consensus on the contents and methods of education to improve recovery knowledge, since few interventional studies targeting for professionals have implemented [43]. Future studies should aim to clarify how recovery knowledge can be increased and contributes to reducing stigma among mental health professionals. Additionally, the SDSJ score among mental health professionals in community facilities was lower than in psychiatric wards. This result implies that working experience in community settings such as outpatient clinics and psychiatric day-care, rather than working in only psychiatric wards, could help to reduce their stigma. On the other hand, the fit of our model was insufficiently high. Therefore, further studies are required to clarify the factors related to stigma among mental health professionals.

The results of the subgroup analyses showed different causal factors of stigma among nurses and psychiatric social workers. Control over practice, educational history, and recovery knowledge were significantly associated with stigma among nurses. However, these variables were not associated with stigma among psychiatric social workers. Unlike other professionals, including psychiatric social workers, nurses spend a great deal of time engaging and taking care of patients so as to support their daily lives throughout the day [44]. Nurses are also likely to experience negative experiences such as refusal of care and aggression by the patients [45,46,47,48]. Such occupational characteristics might influence the factors that cause stigma. Moreover, considering the stressful environments surrounding nurses as mentioned above, control over practice might help to reduce stigma towards people with schizophrenia through improving mental health among nurses [27,49,50]. 

Moreover, as previous studies on psychiatrist and psychiatric nurses have suggested [39,51,52,53], this study also showed that educational history and recovery knowledge are possible factors that can reduce stigma among nurses. Psychiatric social workers had higher levels of recovery knowledge than nurses in the current study. These findings suggest that the causal factors of stigma vary by mental health professionals due to different backgrounds of educational training and philosophies [29,30,31,32]. Interestingly, the explanatory variables we assumed in this study could hardly explain the impacts on stigma towards people with mental illness among psychiatric social workers. Further comprehensive studies among various professionals would further our understanding of these factors. 

This was the first study to examine the association between control over practice in workplace practice environments and stigma towards people with schizophrenia among mental health professionals. Our results offer the implication for managers in mental health professionals that one of the key strategies for reducing stigma among mental health professionals, especially nurses, might be to use intra-organizational resources such as human resources effectively so that mental health professionals can offer better care and enhance control over practice. Future studies should aim to conduct well-designed longitudinal and interventional studies among mental health professionals, especially nurses, in order to obtain insights into the mechanisms and the strategies that control over practice could reduce stigma.

The present study has several limitations. First, due to the cross-sectional design [28,35], the study could not examine the causal association between control over practice and stigma towards people with schizophrenia among mental health professionals. Future studies using longitudinal designs are required to examine the causal associations between these variables. Second, the data was collected nine years ago [28,35], and might not reflect the current situation sufficiently. Given this, our results need to be interpreted carefully. Third, the results of the hierarchical multiple regression analyses might strongly reflect the answers from nurses and psychiatric social workers. Further comprehensive studies among various professionals with sufficient sample size would further our understanding of these factors. Fourth, the valid response rate (58.7%) in our study was not enough high. Some items in the SDSJ scale had missing values in respondents, especially the item related to marriage with people with schizophrenia (i.e., “I would be against any daughter of mine marring a man with schizophrenia who had been in a hospital.” [37]). This could affect the finding in the study. Additionally, social desirability bias could have influenced the results, as pointed out in previous studies [26,51,52,53]. Future studies may consider to use less biased measurement methods, such as the implicit association test for measuring stigma [51,54,55,56]. 

## 5. Conclusions

Control over practice in work environments might help to reduce stigma toward people with schizophrenia among mental health professionals in Japan. Factors associated with stigma might differ by occupation. Therefore, further comprehensive studies among various professionals would further our understanding of these factors.

## Figures and Tables

**Table 1 healthcare-09-00107-t001:** Participants’ occupational and demographic characteristics for total samples, nurses, and psychiatric social workers.

Variables	Total *N* = 279		Nurse *n* = 121 (43.4%)		PSW *n* = 92 (33.0%)	
	*n* [Mean]	(%) [SD]	*n* [Mean]	(%) [SD]	*n* [Mean]	(%) [SD]
**Age (years)**	[40.7]	[11.9]	[45.3]	[1.10]	[35.6]	[1.01]
**Sex (male)**	82	(29.4)	23	(19.2)	32	(34.8)
**Educational history**						
High Education level	152	(54.5)	15	(12.4)	84	(91.3)
Low Education level	127	(45.5)	106	(87.6)	8	(8.7)
**Years of work tenure in psychiatric or mental health services**	[9.9]	[8.5]	[10.8]	[0.80]	[7.89]	[0.67]
**Employment status**						
Full-time job	222	(79.6)	94	(77.7)	78	(84.8)
Part-time job	50	(17.9)	24	(19.8)	12	(13.5)
Unknown	7	(2.5)	3	(2.5)	2	(1.7)
**Settings**						
**Psychiatric Ward**	121	(43.4)	101	(83.5)	3	(3.26)
**Community**						
Out-patient clinic	33	(11.8)	9	(7.44)	8	(8.70)
Psychiatric day-care	26	(9.3)	6	(4.96)	10	(10.9)
Home assistance/rehabilitation	16	(5.7)	2	(1.65)	7	(7.61)
Group home	4	(1.4)	0	(0)	4	(4.35)
Job assistance	60	(21.5)	1	(0.80)	52	(56.5)
Community activity support center	2	(0.7)	0	(0)	2	(2.16)
Home-visit nursing	1	(0.4)	0	(0)	0	(0)
Others	16	(5.7)	2	(1.65)	6	(6.52)
**Recovery knowledge**						
Know well about recovery	146	(52.3)	42	(34.7)	72	(78.3)
Not know much about recovery	108	(38.7)	61	(50.4)	17	(18.5)
Not know about recovery	25	(9.0)	18	(14.9)	3	(3.2)

Note: PSW: Psychiatric social worker.

**Table 2 healthcare-09-00107-t002:** The results of hierarchical multiple linear regression model with forced entry method (*n* = 279).

Variables	Model 1			Model 2			Model 3			Model 4		
	β	SE	*p*	β	SE	*p*	β	SE	*p*	β	SE	*p*
**Control over practice**	−0.22	0.074	<0.001	−0.20	0.071	<0.001	−0.18	0.067	0.001	−0.16	0.066	0.002
Age				0.07	0.017	0.238	0.04	0.016	0.490	0.06	0.016	0.298
Educational history				−0.28	0.410	<0.001	−0.09	0.460	0.206	−0.05	0.449	0.462
**Settings** **(Ward/^a^ Community)**							−0.36	0.451	<0.001	−0.31	0.443	<0.001
**Recovery Knowledge**										−0.24	0.284	<0.001
R	0.222			0.382			0.480			0.529		
R^2^	0.049			0.146			0.230			0.280		
Adjust R^2^	0.046			0.137			0.219			0.267		
SE	3.336			3.173			3.019			2.925		
F	14.33 ***			15.69 ***			20.47 ***			21.21 ***		

Note: The bold characteristics represent statistical significance with *** *p* < 0.001. ^a^ Community: All departments/facilities other than wards. Regarding the settings as the dummy variable, ward was assigned 0 and community was assigned 1, respectively.

**Table 3 healthcare-09-00107-t003:** The results of sub-group analysis among nurses (*n* = 121).

Variables	Model 1			Model 2			Model 3			Model 4		
	β	SE	*p*	β	SE	*p*	β	SE	*p*	β	SE	*p*
**Control over practice**	−0.31	0.106	0.001	−0.35	0.108	<0.001	−0.33	0.108	<0.001	−0.26	0.105	0.003
Age				0.14	0.025	0.140	0.12	0.025	0.204	0.17	0.024	0.052
**Educational history**				0.17	0.894	0.059	0.18	0.889	0.045	0.22	0.851	0.010
Settings (Ward/^a^ Community)							−0.15	0.775	0.099	−0.14	0.735	0.098
**Recovery Knowledge**										−0.32	0.421	<0.001
R	0.308			0.361			0.388			0.493		
R^2^	0.095			0.130			0.150			0.243		
Adjust R^2^	0.087			0.108			0.121			0.210		
SE	3.177			3.140			3.117			2.955		
F	12.44 **			5.83 **			5.13 **			7.37 ***		

Note: The bold characteristics represent statistical significance with ** *p* < 0.01, *** *p* < 0.001. ^a^ Community: All departments/facilities other than wards. Regarding the settings as the dummy variable, ward was assigned 0 and community was assigned 1, respectively.

**Table 4 healthcare-09-00107-t004:** The results of sub-group analysis among psychiatric social workers (*n* = 92).

Variables	Model 1			Model 2			Model 3			Model 4		
	β	SE	*p*	β	SE	*p*	β	SE	*p*	β	SE	*p*
Control over practice	−0.08	0.107	0.457	−0.08	0.108	0.453	−0.07	0.104	0.499	−0.08	0.105	0.450
Age				0.03	0.031	0.767	0.04	0.030	0.685	0.04	0.030	0.731
Educational history				−0.01	1.065	0.898	0.03	1.038	0.742	0.03	1.041	0.750
**Settings** **(Ward/^a^ Community)**							−0.29	1.646	0.006	−0.29	1.651	0.006
Recovery Knowledge										−0.08	0.580	0.444
R	0.079			0.086			0.303			0.313		
R^2^	0.006			0.007			0.092			0.098		
Adjust R^2^	−0.005			−0.026			0.050			0.045		
SE	2.842			2.873			2.764			2.771		
F	0.56			0.22			2.19			1.86		

Note: The bold characteristics represent statistical significance, ^a^ Community: All departments/facilities other than wards. Regarding the settings as the dummy variable, ward was assigned 0 and community was assigned 1, respectively.

## Data Availability

The data that support the findings of this study are available from the corresponding author on request.

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
