# Peer review of "Association between Work Environments and Stigma towards People with Schizophrenia among Mental Health Professionals in Japan"

_healthcare, 2021, doi:10.3390/healthcare9020107_

Round 1
Reviewer 1 Report
Abstract:
Use of the phrase "Workplace Practice Environment" is an odd choice for the matter discussed. Consider rephrasing to work environment or another more appropriate phrase.
Introduction:
Although it is reported later on in the paper inside table 1, consider writing a statement explaining the term "Recovery Knowledge" in the introduction.
Methods:
The data was collected 9 years ago, and may not be representative of the current situation. Considering adding that to the limitations list.
Consider describing the scales used in more details, and further clarifying the parameters being analyzed.
Results:
Kindly explain the various models used in table 2
Discussion:
This appears to be very overlapping the Result section. Consider entering statistical information in the result section, and not merely repeating the results in the discussion section, but using this section to explain and "discuss" the results. This has been done in some statements, but can be further improved.
Reviewer 2 Report
This is, in summary, an interesting paper aimed to examine the association between control over practice in workplace practice environments and stigma toward people with schizophrenia among mental health professionals. The authors reported that control over practice was negatively associated with stigma among mental health professionals. Moreover, the subgroup analyses among nurses indicated that control over practice, educational history and recovery knowledge were associated with stigma. However, these variables were not associated with stigma among psychiatric social workers. Finally, control over practice might help to reduce stigma among mental health professionals and factors related to stigma might differ by occupation.
The authors may find as follows my main comments/suggestions.
First, when the authors throughout the Introduction section correctly focused on the impact of stigma in disabling psychiatric conditions such as schizophrenia, they could further stress the the determinants of stigmaization in this invalidating disorder. In fact, genetic explanation of schizophrenia may potentially enhance stigma. In particular, considering schizophrenia as a genetic disorder influenced participants perception of other people's beliefs about dangerousness and unpredictability and people's desire for social distance. Importantly, a genetic explanation of schizophrenia was more frequently associated with stigmatizing attitudes. According to a study which has been published in 2013 on J Psychiatr Ment Health Nurs (PMID: 21848591), there were high levels of perceived stigmatization in medical students and medical doctors and at least half of the analyzed subjects perceived stigmatizing social attitudes against psychotic individuals. Thus, given the above information, my additional suggestion is also to rapidly include, throughout the present manuscript, the mentioned paper (PMID: 21848591). Moreover, the genetic liability of disabling conditions like schizophrenia needs to be enphasized. In particular, according to a study which has been published in 2014 on World J Biol Psychiatry, gray matter reductions in the anterior cingulate have been reported as markers of genetic liability to psychosis, while reductions in the superior temporal gyrus and cerebellum may be interpreted as markers of a first onset of the illness. Thus, i suggest to briefly cite, within the main text, the specified paper on this specific topic (PMID: 22283467).
In addition, as the main aims/objectives of this study has been described by the authors, the most relevant hypotheses of this study should be similarly and extensively reported within the main text.
Importantly, the main reasons underlying the lower (58.7%) valid response rate need to be specified. Here, the authors should provide their specific comments to this specific regard (e.g., within the main limitations/shortcomings of the present paper).
Furthermore, the authors should immediately present the main findings of this paper in the first lines of the Discussion section instead of focusing on the most relevant study strengths of this manuscript that should be stressed elsewhere within the main text.
Finally, what is the take-home message of this manuscript? While the authors reported that, control over practice in workplace practice environments might help to reduce stigma toward people with schizophrenia among mental health professionals and factors associated with stigma might differ by occupation, they failed, in my opinion, to provide the most relevant cocnlusive remarks of their paper. How stigma toward people with schizophrenia might be really attenuated? How effective strategies aimed to reduce stigmatization in schizophrenia might be conducted? Here, more details/ information are needed.
